# Accuracy measurement of different marker based motion analysis systems for biomechanical applications: A round robin study

Stefan Schroeder[1], Sebastian Jaeger[1], Jonas Schwer[2], Andreas Martin Seitz[2], Isabell Hamann[3], Michael Werner[3], Christoph Thorwaechter[4], Inês Santos[4], Toni Wendler[5], Dennis Nebel[6], Bastian Welke[6]*

1 Laboratory of Biomechanics and Implant Research, Department of Orthopaedics, Heidelberg University Hospital, Heidelberg, Germany, 2 Institute of Orthopaedic Research and Biomechanics, Centre for Trauma Research Ulm, Ulm University Medical Centre, Ulm, Germany, 3 Fraunhofer Institute for Machine Tools and Forming Technology, Dresden, Germany, 4 Department of Orthopaedics and Trauma Surgery, Musculoskeletal University Center Munich (MUM), University Hospital, LMU Munich, Munich, Germany, 5 ZESBO—Center for Research on Musculoskeletal Systems, Leipzig University, Leipzig, Germany, 6 Laboratory for Biomechanics and Biomaterials, Department of Orthopaedic Surgery, Hannover Medical School, Hannover, Germany

☯ These authors contributed equally to this work.
* welke.bastian@mh-hannover.de

**Data Availability Statement:** All relevant data are within the paper and its Supporting Information files.

## Abstract

### Introduction

Multiple camera systems are widely used for 3D-motion analysis. Due to increasing accuracies these camera systems gained interest in biomechanical research areas, where high precision measurements are desirable. In the current study different measurement systems were compared regarding their measurement accuracy.

### Materials and methods

Translational and rotational accuracy measurements as well as the zero offset measurements of seven different measurement systems were performed using two reference devices and two different evaluation algorithms. All measurements were performed in the same room with constant temperature at the same laboratory. Equal positions were measured with the systems according to a standardized protocol. Measurement errors were determined and compared.

### Results

The highest measurement errors were seen for a measurement system using active ultrasonic markers, followed by another active marker measurement system (infrared) having measurement errors up to several hundred micrometers. The highest accuracies were achieved by three stereo camera systems, using passive 2D marker points having errors typically below 20 μm.

**Funding:** The author(s) received no specific funding for this work. Publication costs are covered by the German Research Foundation (DFG) and the Open Access Publication Fund of Hannover Medical School (MHH).

**Competing interests:** [I have read the journal's policy and the authors of this manuscript have the following competing interests: [Sebastian Jaeger reports grants from B Braun Aesculap, Johnson & Johnson Depuy Synthes, Heraeus Medical, Waldemar Link, Peter Brehm, Ceramtec, Implantcast, Mathys Orthopaedie GmbH and Zimmer Biomet that are not related to the current study. This does not alter our adherence to PLOS ONE policies on sharing data and materials."]

## Conclusions

This study can help to better assess the results obtained with different measurement systems. With the focus on the measurement accuracy, only one aspect in the selection of a system was considered. Depending on the requirements of the user, other factors like measurement frequency, the maximum analyzable volume, the marker type or the costs are important factors as well.

## Introduction

Marker based motion capture analysis is a common approach to make three dimensional human motions visible. Research areas, in which motion analysis is applied, include the analysis and optimization of training methods in sports as well as the examination of human motion for health reasons [1–3]. For these applications, the markers are attached to the skin of human subjects on top of bony landmarks and the joint angles during motion can be calculated with different models. Because the skin with the attached markers is moving relative to the anatomical landmarks during motion and a model to determine the joint centers always contain errors, resulting measurement errors of a few millimeters and a few degrees are unavoidable [2, 4, 5]. Algorithms were developed to limit errors due to soft tissue motion and increase the measurement accuracy [6]. However, in the mentioned research fields, typically the measurement accuracy does not need to be beyond one millimeter or one degree. In biomechanical studies with human specimens, where the kinematics of an isolated joint is analyzed during passive motion, a higher precision of the motion analysis is needed. Small differences of the joint kinematics can help to optimize implant designs or surgical methods. For biomechanical studies, optical markers can directly be attached to the bone using pins building a rigid unit [7–9]. Therefore, measurement artefacts as when markers are attached at the soft tissue can be prevented, leading to a higher accuracy [4]. Another biomechanical research field next to the kinematics of the joint is the measurement of micromotions between an artificial joint and a bone of a human specimen, in order to determine the primary stability of the implant [10, 11]. For this purpose, markers are attached to the implant as well as to the bone to measure the micromotions between the components under loading conditions. The translational motions between the implant and the bone are often below 100 μm and rotations far below one degree can occur [12–14]. Due to improvements of marker-based measurement systems in the last decades, these systems were also used for these biomechanical research areas, where high accuracies are needed.

When acquiring a new optical measurement system for minor motions, the accuracy in all six degrees of freedom of the system is important. The manufacturers of a measurement system typically state either no accuracy or an accuracy, which mostly refers to an in-house measurement of the company in one axis under optimum conditions. Therefore, the aim of this round robin test was to compare the measurement accuracies of different available motion capture systems, when experienced users carry out the same measuring task. The results can help to detect possible variances of the measurement systems, make studies more comparable and should give scientists an overview of the accuracy of some commonly used marker based measurement systems.

## Materials and methods

In total, seven different measurement systems from six biomechanical laboratories located in Germany were tested regarding the measurement accuracy and the precision. For this end,

two different reference devices were used. All measurements were performed in the same standardized temperature-controlled (22 ± 1°C) precision measurement room.

## Camera systems

The used camera systems and the abbreviation for this publication are: NDI Optotrak Certus (Northern Digital Inc., Waterloo, Ontario, Canada) = Optotrak, CMS20S-2-Sync (Zebris Medical GmbH, Isny, Germany) = CMS, Q-400-3D (Limess Messtechnik und Software GmbH, Krefeld, Germany) = Q-400, Pontos 5M (GOM GmbH, Braunschweig, Germany) = Pontos, OptiTrack (NaturalPoint, Inc., Corvallis, Oregon, USA) = OptiTrack, Atos Core 300 (GOM GmbH, Braunschweig, Germany) = Core and Aramis 3D camera MV600 (GOM GmbH, Braunschweig, Germany) = Aramis. Fig 1 shows the seven optical measurement systems used in this study and Table 1 contains the properties of the measurement systems including the type of markers and the measurement resolution according to the manufacturer.

The measurements of the different systems were performed at different time points and, with one exception, on different days. The measuring systems were placed in the temperature controlled precision measuring room at least one hour before the measurement for acclimatization. Afterwards, the responsible working group or researcher of the particular measurement

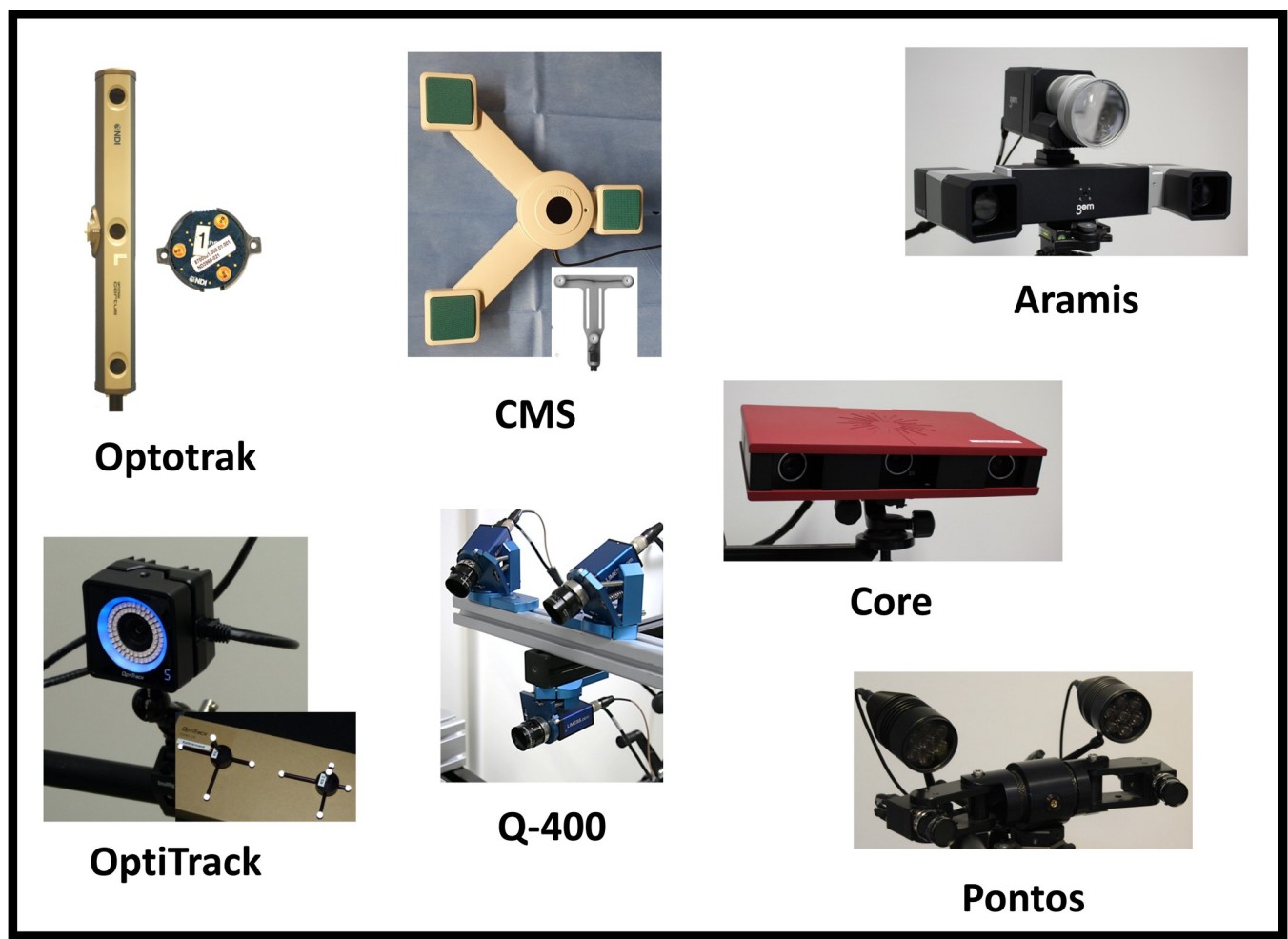

**Fig 1. Images of the seven measurement systems used for the accuracy and zero offset measurements.**

**Table 1. Specifications of the seven measurement systems.**

| Measurement System | Markers | Measurement Accuracy/Resolution (Manufacturer) | Used Frequency/ Max. Frequency | Max. analyzable volume | Calibration | Camera |
|---|---|---|---|---|---|---|
| Optotrak | Active markers, Orthopaedic Research Pins (20mm) | 0.1 mm/ 0.01 mm | 410/2000 Hz | 4.2x3.0x5.5 m | Calibration was carried out by the manufacturer | Three camera sensors |
| CMS | Active Ultrasonic markers; 6 active transmitters | 1/10 mm– 1/100 mm | Max. 300 Hz/ number of marker | Hemisphere of 16.75 m$^3$ (max. distance = 2 m) | Calibration was carried out by the manufacturer | Ultrasonic receiver MA-XX-2 |
| Q-400 | Passive marker (Speckle-pattern) | 0.01 pixel for 3D-motions | 15 Hz | Up to 10x10 m; low depth measurement | Special calibration targets used before measurement | Three cameras with 2.0 Mpixel |
| Pontos | Passive markers 1.5 mm (GOM) | 0.021 pixel (calibration error) | 15 Hz | 280x240x240 mm | Calibration object 20 MV 250x200mm$^2$ | Stereo camera system (two 3 Mpixel cameras) |
|  |  | No company details |  |  |  |  |
| OptiTrack | Passive Markers; two rigid bodies KS1 = 5 markers & KS2 = 4 markers | +- 0.2 mm | 240/240 Hz | 5x5x3 m | Using a calibration stick with passive markers on it before measurement | Seven cameras (Prime 13) |
|  |  | 1.3 MP |  |  |  |  |
| Core | Passive markers 1.5 mm (GOM) | 0.008 pixel (optimized calibration error) | 7/14 Hz | 300x230x230 mm | Calibration object CP40/ MV320 | Stereo camera system (two 5 Mpixel cameras) |
|  |  | No company details |  |  |  |  |
| Aramis | Passive markers 3.0 mm (GOM) | 0.015 pixel (optimized calibration error) | 25/44 Hz | 600x530x400 mm | Calibration object CP40/ MV560 | Stereo camera system (two 6 Mpixel cameras) |
|  |  | No company details |  |  |  |  |

system set up the measurement system as they usually do for laboratory tests. Depending on the measurement system, a calibration was performed before the measurement. After that, the measurements were performed with both reference devices successively.

## Reference devices and measurement sequences

Two different reference devices were used, in order to ensure the accuracy of the test results and to cover two different measuring ranges. In addition, the marker motion of the first reference system was measured in relation to another fixed marker and the marker motion of the second reference system was determined in relation to the same marker in the zero position.

**Coordinate measurement machine.** The coordinate measurement machine (CMM, MS222, Mahr, Göttingen, Germany) is positioned in the corner of the measurement room and has an accuracy of 2.0 μm. The size of the measurement room is 17 m$^2$ floor space and 2.5 m height. The baseplate of the CMM can move in the x- and z-direction, whereby a touch sensor holder can move in the y-direction. The CMM was used for the accuracy measurement of greater distances (0.1 to 100 mm) and to determine the zero offset of the measurement devices.

Two adapters, having a flat rectangular surface, were mounted to the CMM, used as attachment for the different markers. One adapter was fixed at the baseplate and the other one at the touch sensor holder. To measure the accuracy as well as the zero offset of the x-axis, y-axis and z-axis, six movement routines were executed successively. Every measurement started at the home position. In the home position, the adapter for the y-axis was just above the adapter for the x-axis and z-axis (Fig 2). In order to reduce vibrations, the feet of the CMM are made of elastic rubber.

To determine the zero offset (1) a measurement was performed at the home position, then (2) the adapter was driven 10 mm in one axial direction, (3) a measurement was performed again and afterwards (4) the adapter was driven back to the home position and (5) another

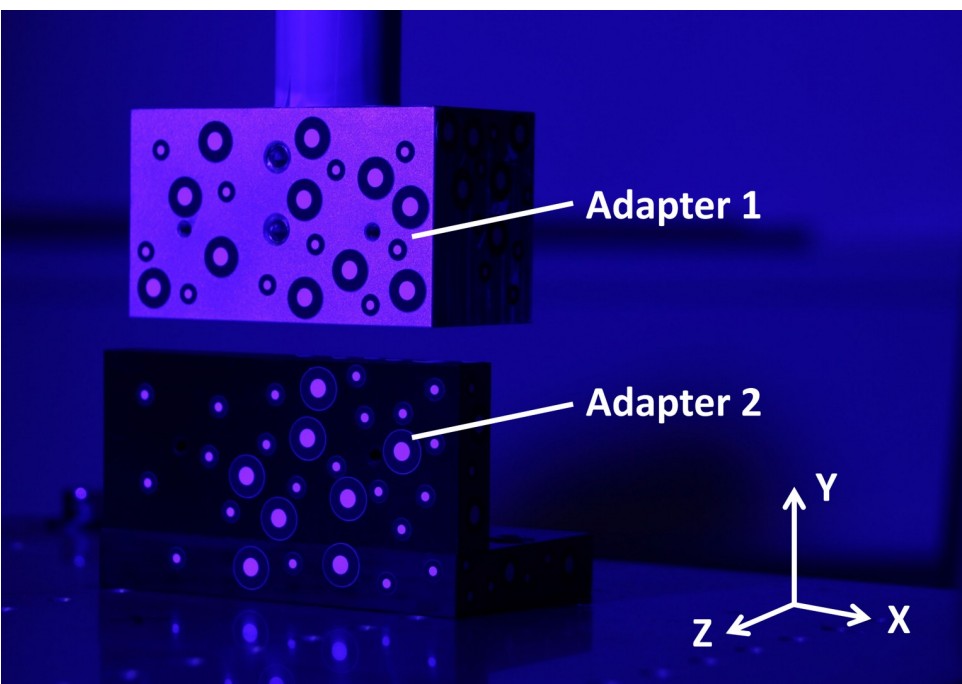

**Fig 2. Adapters mounted at the CMM in order to determine the measurement accuracy and zero offset.**

measurement was performed. This process was repeated five times for statistical purposes. After that, the same protocol was applied for the other two axes. Thus, for each axis, six measurements were performed at zero position and five measurements were performed at a distance of 10 mm. The distances from the moving adapter were determined in relation to the markers of the motionless adapter and compared to the real motion of the CMM to calculate the measurement errors.

In order to measure the accuracy for different distances, the same approach was used as for the previous measurement, but the three axes were driven successively to different positions from the starting position (Table 2) and this procedure was measured five times for statistical purposes.

After each position, a measurement was performed. Using this method, the measurement errors could be determined from very low motions (0.1 mm) to larger distances (100 mm).

**Manual reference device.** The manual reference device is a six degree of freedom adjustment unit, consisting of three linear bearing stages M-443 including the matching micrometer screws SM-50 (Newport Corporation, Irvine, California, USA) having a measurement sensitivity of 1 μm each and three different rotating bearings. The rotating bearings were Newport M-GON40-L for rotations around the x-axis (sensitivity of 5 arcsec), Newport M-GON40-U for rotations around the y-axis (sensitivity of 8 arcsec) and Newport RS-65 for rotations around the z-axis (sensitivity of 11 arcsec).

The unit can move in one direction or around one axis by drilling a specific micrometer screw (Fig 3).

**Table 2. Positions for the translational accuracy measurement using the CMM.**

|  | P0 | P1 | P2 | P3 | P4 | P5 | P6 | P7 | P8 | P9 |
|---|---|---|---|---|---|---|---|---|---|---|
| Position in mm | 0.000 | 0.100 | 1.000 | 3.000 | 5.000 | 10.000 | 30.000 | 50.000 | 70.000 | 100.000 |

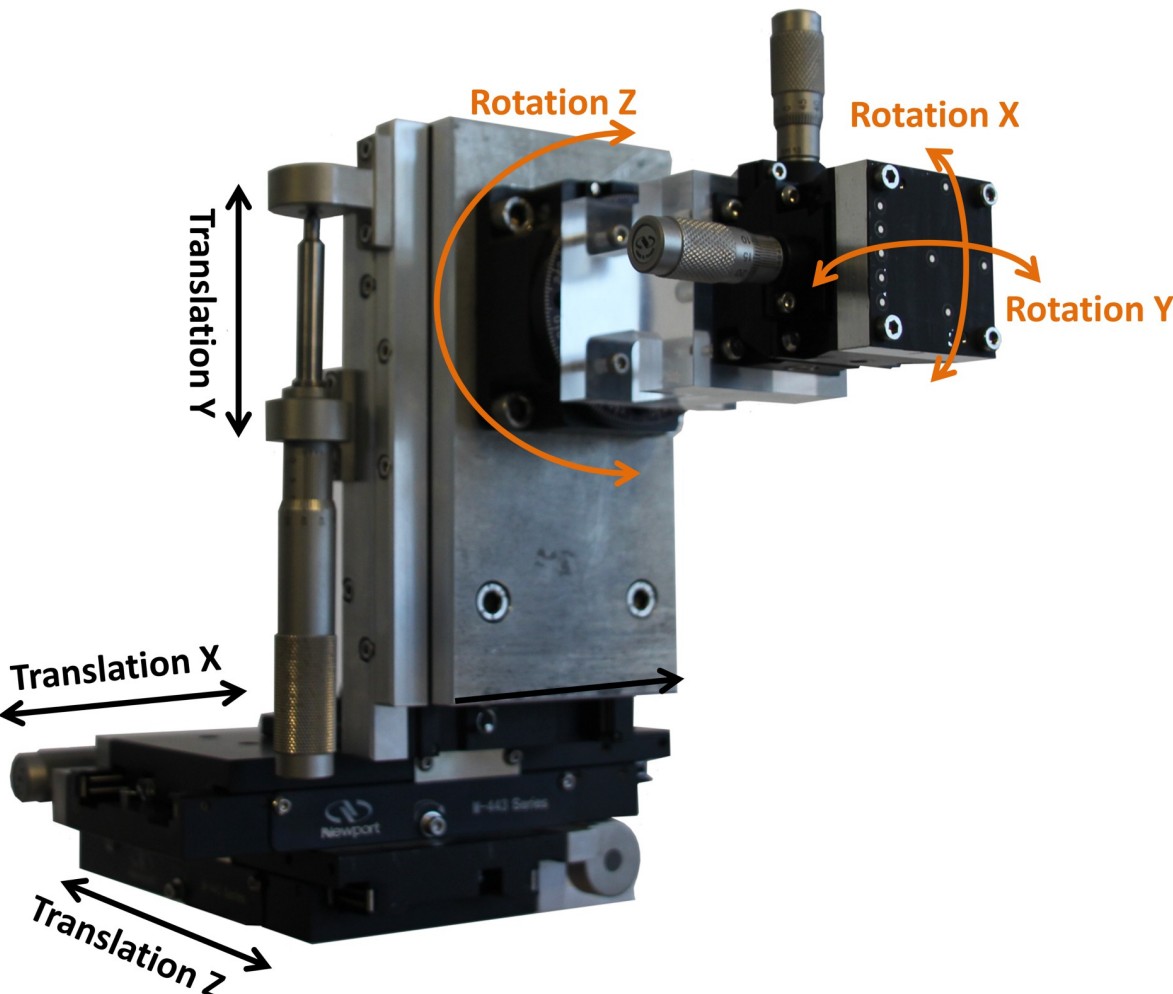

**Fig 3. Manual reference device to determine the accuracy of small translational as well as rotational motions.**

The manual reference device was fixated at a stable metal frame, which was positioned next to the CMM in the same measurement room. The different markers were fixated on the front plate of the reference device. To calculate the measurement errors, the distances of the markers on the moving reference device were determined relative to the same markers in the zero positions. Therefore, no reference markers on any motionless part were needed.

The six axes were driven successively to different positions by the same user (Table 3).

The given values of Table 3 correspond to full rotations of the micrometer screws in order to reduce possible user depending variabilities. Measurements were taken at each of the indicated positions. The measurements were repeated five times for statistical purposes.

**Table 3. Translational and rotational positions using the manual reference device.**

|  | P0 | P1 | P2 | P3 |
|---|---|---|---|---|
| Translation in mm (x, y, z) | 0.000 | 1.000 | 3.000 | 5.000 |
| Rotation x in˚ | 0.000 | 1.242 | 3.727 | 6.212 |
| Rotation y in˚ | 0.000 | 2.116 | 6.349 | 10.581 |
| Rotation z in˚ | 0.000 | 2.000 | 6.000 | 10.000 |

## Statistics

The measured values with the measurement devices were compared with the values of the CMM and with the values of the manual adjustment unit to determine the measurement errors with both reference devices. The mean and standard deviation of the five measurements were calculated for every position. All statistical analyses were carried out using SPSS 22 (IBM, Amonk, NY, USA).

## Results

Some of the measurement systems could determine motions in one direction (Pontos, Core and Aramis) by defining a coordinate system in the associated software. The Optotrak and OptiTrack systems used a rigid reference marker body to define the coordinate system. On the other hand, the two measurement devices Q-400 and CMS measured a resulting vector motion from x-, y- and z-translation due to the usage of a global coordinate system. In order to compare the measurement values of the different measurement devices all together, the resulting measured motions of all axes were used and compared to the true values. Because of high differences regarding the measurement errors a logarithmic scale was used for all figures without the presentation of the standard error.

### Zero offset measurement results using the coordinate measurement machine

The measurement errors of the zero offset measurements are shown in Fig 4. Zero_X represents five measurements when driving from 10 mm in x-direction to the home position. P_X represents five measurements when driving from the home position 10 mm in x-direction. Zero_Y, P_Y, Zero_Z and P_Z represent similar measurements like Zero_X and P_X but for movements in y- and z-direction.

The total values of the zero offset measurement errors and the associated standard deviation of the zero offset measurements are shown in Table 4.

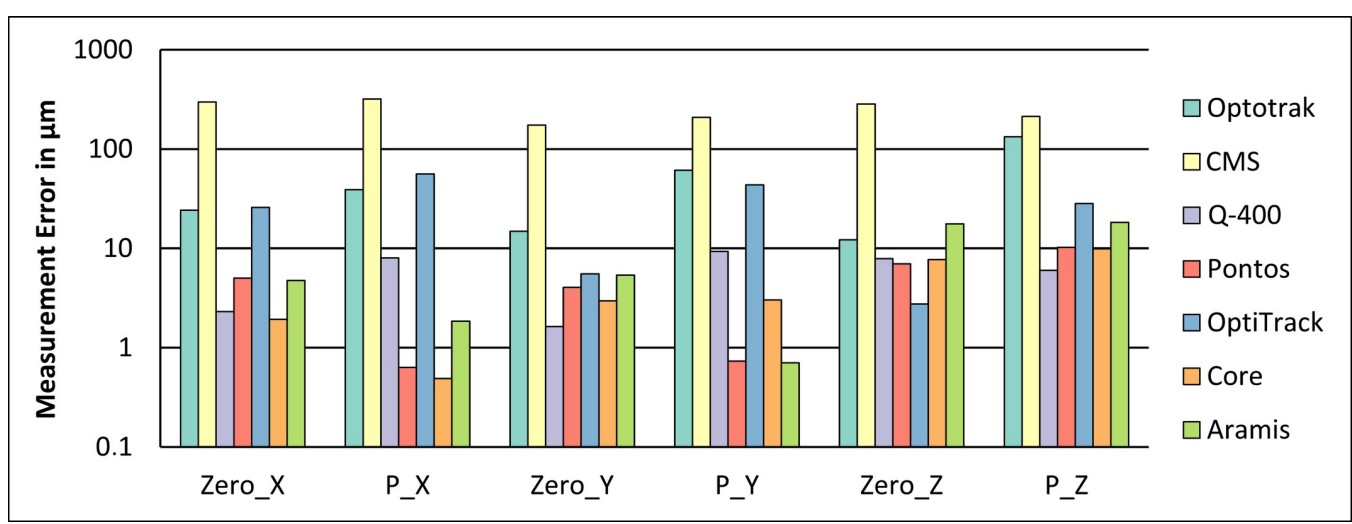

**Fig 4. Measurement errors for the zero offset condition in x-, y- and z-direction of the investigated measurement systems.**

**Table 4. Mean measurement errors and standard deviation of the zero offset condition of the seven systems.**

|  | Optotrak | CMS | Q-400 | Pontos | OptiTrack | Core | Aramis |
|---|---|---|---|---|---|---|---|
| Zero_X in µm | 24 ± 11 | 298 ± 196 | 2 ± 2 | 5 ± 2 | 26 ± 25 | 2 ± 1 | 5 ± 3 |
| P_X in µm | 39 ± 4 | 319 ± 214 | 8 ± 1 | 1 ± 0 | 56 ± 1 | 1 ± 0 | 2 ± 1 |
| Zero_Y in µm | 15 ± 7 | 174 ± 125 | 2 ± 1 | 4 ± 1 | 6 ± 8 | 3 ± 1 | 5 ± 2 |
| P_Y in µm | 61 ± 1 | 208 ± 95 | 9 ± 0 | 1 ± 1 | 44 ± 7 | 3 ± 1 | 1 ± 1 |
| Zero_Z in µm | 12 ± 3 | 284 ± 193 | 8 ± 4 | 7 ± 3 | 3 ± 3 | 8 ± 3 | 18 ± 8 |
| P_Z in µm | 133 ± 3 | 213 ± 103 | 6 ± 5 | 10 ± 3 | 28 ± 21 | 10 ± 3 | 18 ± 5 |

## Accuracy measurement results using the coordinate measurement machine

The measurement errors of the accuracy measurements in x-, y- and z-direction using the CMM are shown in Fig 5. The target point P6 of the measurement in z-direction for the Opti-Track measurement system was deleted because one marker of the rigid body was covered and could not be detected.

The total values of measurement errors of the accuracy measurements in x-, y- and z-direction using the CMM are shown in Table 5.

## Accuracy measurement using the manual reference device

The results of the measurements using the manual adjustment unit are separated into translational and rotational results. All adjustments of the manual device for each group were performed by the same experienced user to account for interpersonal variability.

**Translational accuracy measurement using the manual reference device.** The measurement errors of the translational accuracy measurements in x-, y- and z-direction using the manual measurement device are shown in Fig 6.

The total values of the measurement errors and the associated standard deviation of the translational accuracy measurements in x-, y- and z-direction using the manual measurement device are shown in Table 6.

**Rotational accuracy measurement using the manual measurement device.** The measurement errors of the rotational accuracy measurements around the x-, y- and z-axes using the manual measurement device are shown in Fig 7.

The total values of the measurement errors and the associated standard deviation of the rotational accuracy measurements around the x-, y- and z-axes using the manual measurement device are shown in Table 7.

## Discussion

In the current study seven different optical measurement systems were compared regarding their accuracies and zero offset in a round robin test. The measurement systems can be divided into two groups: the ones using active markers, and those using passive reflective markers (Q-400, Pontos, OptiTrack, Core and Aramis). The systems with active markers use either ultrasonic (CMS) or optical infrared (Optotrak) markers. The Optotrak system can be equipped with up to eight position sensors. The measurement systems using passive markers can also be divided in two subgroups. Three systems are stereo camera systems from the same manufacturer and use round passive marker dots (Pontos, Core and Aramis). Two systems can be equipped with multiple cameras, whereby one system uses rigid bodies with spherical markers (OptiTrack) and one uses a speckle pattern for motion detection (Q-400).

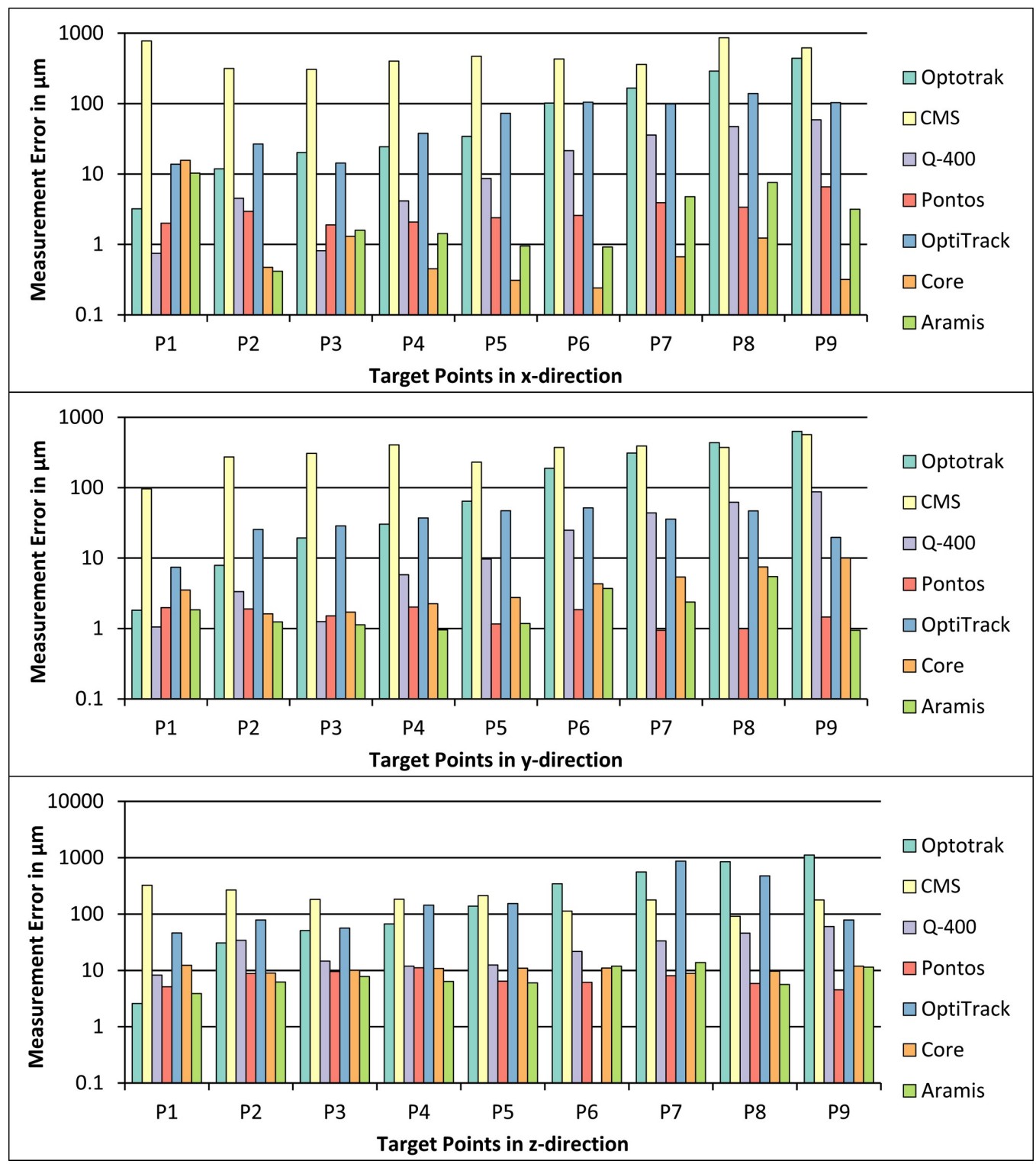

**Fig 5.** Measurement errors of the accuracy measurements of translational motions in x-direction (top diagram), y-direction (middle diagram) and z-direction (bottom diagram) for the seven measurement systems using the CMM as reference device.

**Table 5. Numerical measurement errors of the seven systems when using the CMM as reference device.**

|  | Optotrak | CMS | Q-400 | Pontos | OptiTrack | Core | Aramis |
|---|---|---|---|---|---|---|---|
| P1_X in µm | 3 ± 2 | 779 ± 573 | 1 ± 1 | 2 ± 1 | 14 ± 7 | 16 ± 0 | 10 ± 4 |
| P2_X in µm | 12 ± 3 | 316 ± 231 | 5 ± 1 | 3 ± 3 | 27 ± 11 | 1 ± 0 | 0 ± 0 |
| P3_X in µm | 20 ± 2 | 306 ± 224 | 1 ± 1 | 2 ± 1 | 14 ± 12 | 1 ± 0 | 2 ± 1 |
| P4_X in µm | 24 ± 3 | 401 ± 185 | 4 ± 1 | 2 ± 2 | 38 ± 11 | 1 ± 0 | 1 ± 1 |
| P5_X in µm | 34 ± 1 | 472 ± 206 | 9 ± 0 | 2 ± 2 | 73 ± 13 | 0 ± 0 | 1 ± 1 |
| P6_X in µm | 102 ± 3 | 431 ± 545 | 21 ± 1 | 3 ± 2 | 105 ± 13 | 0 ± 0 | 1 ± 1 |
| P7_X in µm | 167 ± 2 | 361 ± 251 | 36 ± 1 | 4 ± 2 | 99 ± 17 | 1 ± 1 | 5 ± 2 |
| P8_X in µm | 290 ± 2 | 861 ± 546 | 47 ± 1 | 3 ± 2 | 139 ± 13 | 1 ± 1 | 8 ± 1 |
| P9_X in µm | 442 ± 4 | 620 ± 257 | 59 ± 2 | 7 ± 3 | 103 ± 12 | 0 ± 0 | 3 ± 0 |
| P1_Y in µm | 2 ± 1 | 96 ± 85 | 1 ± 1 | 2 ± 1 | 7 ± 5 | 4 ± 1 | 2 ± 1 |
| P2_Y in µm | 8 ± 3 | 274 ± 152 | 3 ± 1 | 2 ± 1 | 26 ± 9 | 2 ± 1 | 1 ± 1 |
| P3_Y in µm | 19 ± 2 | 308 ± 292 | 1 ± 0 | 2 ± 0 | 29 ± 8 | 2 ± 1 | 1 ± 1 |
| P4_Y in µm | 30 ± 1 | 407 ± 132 | 6 ± 2 | 2 ± 1 | 37 ± 7 | 2 ± 2 | 1 ± 1 |
| P5_Y in µm | 64 ± 3 | 231 ± 162 | 10 ± 1 | 1 ± 1 | 47 ± 10 | 3 ± 2 | 1 ± 1 |
| P6_Y in µm | 189 ± 2 | 374 ± 417 | 25 ± 2 | 2 ± 1 | 52 ± 7 | 4 ± 2 | 4 ± 1 |
| P7_Y in µm | 311 ± 3 | 391 ± 263 | 44 ± 2 | 1 ± 1 | 36 ± 18 | 5 ± 3 | 2 ± 2 |
| P8_Y in µm | 437 ± 3 | 374 ± 237 | 62 ± 3 | 1 ± 1 | 47 ± 27 | 8 ± 3 | 6 ± 2 |
| P9_Y in µm | 630 ± 2 | 569 ± 348 | 87 ± 4 | 2 ± 1 | 20 ± 30 | 10 ± 3 | 1 ± 1 |
| P1_Z in µm | 3 ± 1 | 325 ± 151 | 8 ± 8 | 5 ± 4 | 46 ± 43 | 12 ± 11 | 4 ± 2 |
| P2_Z in µm | 31 ± 5 | 268 ± 244 | 34 ± 20 | 9 ± 5 | 78 ± 58 | 9 ± 8 | 6 ± 5 |
| P3_Z in µm | 51 ± 4 | 183 ± 106 | 15 ± 14 | 10 ± 5 | 56 ± 45 | 10 ± 9 | 8 ± 6 |
| P4_Z in µm | 67 ± 5 | 184 ± 130 | 12 ± 11 | 11 ± 7 | 145 ± 107 | 11 ± 8 | 6 ± 1 |
| P5_Z in µm | 138 ± 8 | 212 ± 162 | 13 ± 12 | 6 ± 3 | 154 ± 68 | 11 ± 7 | 6 ± 6 |
| P6_Z in µm | 344 ± 6 | 113 ± 66 | 22 ± 14 | 6 ± 4 | - | 11 ± 12 | 12 ± 7 |
| P7_Z in µm | 560 ± 12 | 179 ± 100 | 33 ± 21 | 8 ± 8 | 870 ± 83 | 9 ± 8 | 14 ± 2 |
| P8_Z in µm | 852 ± 18 | 92 ± 138 | 46 ± 14 | 6 ± 2 | 477 ± 164 | 10 ± 10 | 6 ± 5 |
| P9_Z in µm | 1113 ± 21 | 178 ± 86 | 60 ± 20 | 5 ± 5 | 79 ± 72 | 12 ± 10 | 11 ± 10 |

## Coordinate measurement machine

The results of the zero offset measurements revealed the highest measurement errors for the CMS system for all three directions (mostly above 150 µm). The Optotrak and the OptiTrack system measurement errors were above 100 µm when driving to the position P_Z in z-direction. All other systems showed measurement errors below 20 µm for all directions.

Similar results could be seen for the step-wise translational measurements in the three directions. Except of one position in y-direction (P1) and one position in z-direction (P8), the CMS system had measurement errors of more than 100 µm. The Optotrak system clearly shows an increase in measurement error when increasing the distance, ending with the highest measurement error for P8 and P9 in y-direction (>400 µm) and z-direction (>800 µm) of all measurement systems. The OptiTrack system shows the highest measurement errors of the passive marker systems, except for P1 in x-direction and P7, P8 and P9 in y-direction. In addition, the OptiTrack system shows the highest measurement error of all systems in the z-direction for P7 (>700 µm). The Q-400 system showed, especially for small movements (P1 to P5), good measurement accuracies with errors below 20 µm in all directions. Longer distances resulted in higher measurement errors ending in errors between 59 and 87 µm for P9 in the three directions. Both systems using multiple cameras (OptiTrack and Q-400) showed a trend

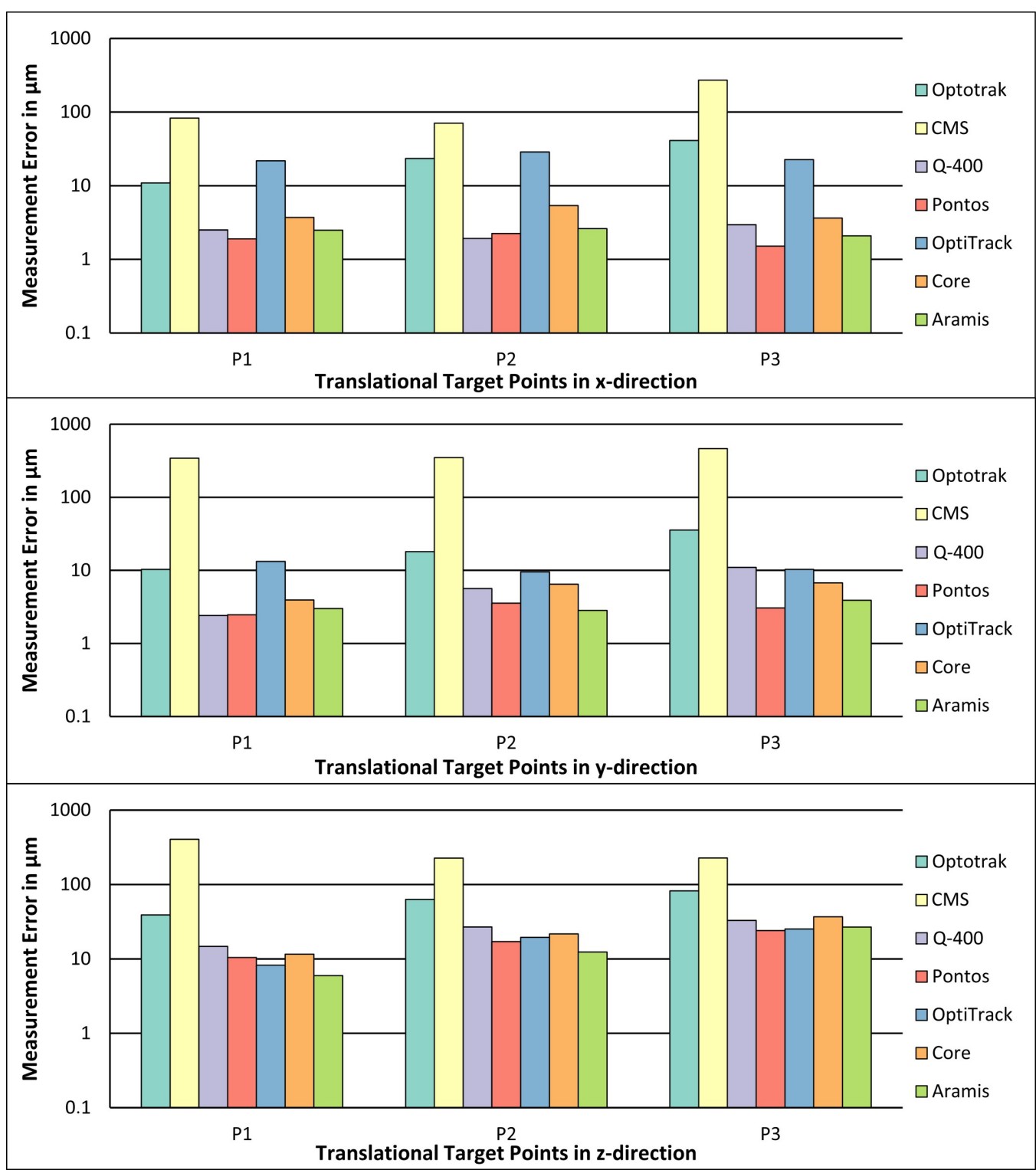

**Fig 6.** Measurement errors of the accuracy measurements of the translational motions in x-direction (top diagram), y-direction (middle diagram) and z-direction (bottom diagram) for the seven measurement systems using the manual reference device.

**Table 6. Numerical translational measurement errors of the seven systems when using the manual reference device.**

|  | Optotrak | CMS | Q-400 | Pontos | OptiTrack | Core | Aramis |
|---|---|---|---|---|---|---|---|
| P1_X in µm | 11 ± 6 | 83 ± 63 | 3 ± 2 | 2 ± 2 | 22 ± 26 | 4 ± 3 | 3 ± 1 |
| P2_X in µm | 23 ± 9 | 71 ± 54 | 2 ± 1 | 2 ± 1 | 29 ± 28 | 5 ± 3 | 3 ± 2 |
| P3_X in µm | 41 ± 8 | 271 ± 296 | 3 ± 2 | 2 ± 1 | 23 ± 26 | 4 ± 2 | 2 ± 2 |
| P1_Y in µm | 10 ± 6 | 343 ± 136 | 2 ± 2 | 3 ± 1 | 13 ± 5 | 4 ± 2 | 3 ± 3 |
| P2_Y in µm | 18 ± 11 | 348 ± 237 | 6 ± 2 | 4 ± 1 | 10 ± 6 | 7 ± 4 | 3 ± 3 |
| P3_Y in µm | 36 ± 8 | 463 ± 333 | 11 ± 3 | 3 ± 2 | 10 ± 6 | 7 ± 3 | 4 ± 2 |
| P1_Z in µm | 39 ± 18 | 404 ± 195 | 15 ± 12 | 10 ± 8 | 8 ± 9 | 12 ± 8 | 6 ± 6 |
| P2_Z in µm | 63 ± 16 | 226 ± 150 | 27 ± 13 | 17 ± 12 | 20 ± 10 | 22 ± 7 | 12 ± 6 |
| P3_Z in µm | 82 ± 14 | 227 ± 117 | 33 ± 16 | 24 ± 8 | 25 ± 12 | 37 ± 8 | 27 ± 9 |

of an increasing measurement error, when increasing the distance. The measurement errors of the Pontos, the Core and the Aramis systems were below 20 µm for all measurement positions in all three directions. The three stereo camera systems and the CMS system did not show an increasing measurement error with an increasing distance. These results show that the stereo camera systems had the highest measurement accuracies for the zero offset and translational motions up to 100 mm. For longer distances however, no statements can be made.

## Manual reference device

The translational measurement errors of the CMS system were the highest when using the manual reference device, having errors of more than 70 µm for all positions in x-direction and errors of more than 200 µm for every position in the other two directions. The second highest measurement errors were detected by the Optotrak system with measurement errors between 10 µm and 82 µm for all positions of the three directions, whereby an increased error for an increased distance can be seen as for the measurements using the CMM. All measurement systems using the passive markers showed measurement errors below 20 µm for the translation in y-direction. For the translation in x-direction, all passive marker systems had measurement errors below 20 µm, except for the OptiTrack system, which showed measurement errors between 20 µm and 30 µm. For the translation in z-direction, all passive marker systems had higher measurement errors than for the other directions, but still below 40 µm. Thus, for very small translational motions, the three stereo camera systems and the Q-400 System showed the lowest measurement errors.

The rotational measurement errors were also the highest for the CMS system for rotations around the y-axis and z-axis. The highest rotational measurement errors for motions around the x-axis were seen for the OptiTrack system followed by the CMS system. The rotational measurement errors for the three stereo camera systems were lowest, not exceeding 5 arcmin. For biomechanical applications, where typically translations and rotations occur, the translational and rotational measurement errors could sum up, which may lead in even higher total measurement errors.

## Comparison of the two reference systems

For the CMM as reference system, the distance of the moving markers was compared relative to a fixed marker. For the manual reference system, the distance of the moving markers was determined relative to the same markers in the zero position. Thus, two different evaluation algorithms were used. The CMM is a user independent and reproducible device, which makes it a suitable tool to determine measurement errors of different measurement systems.

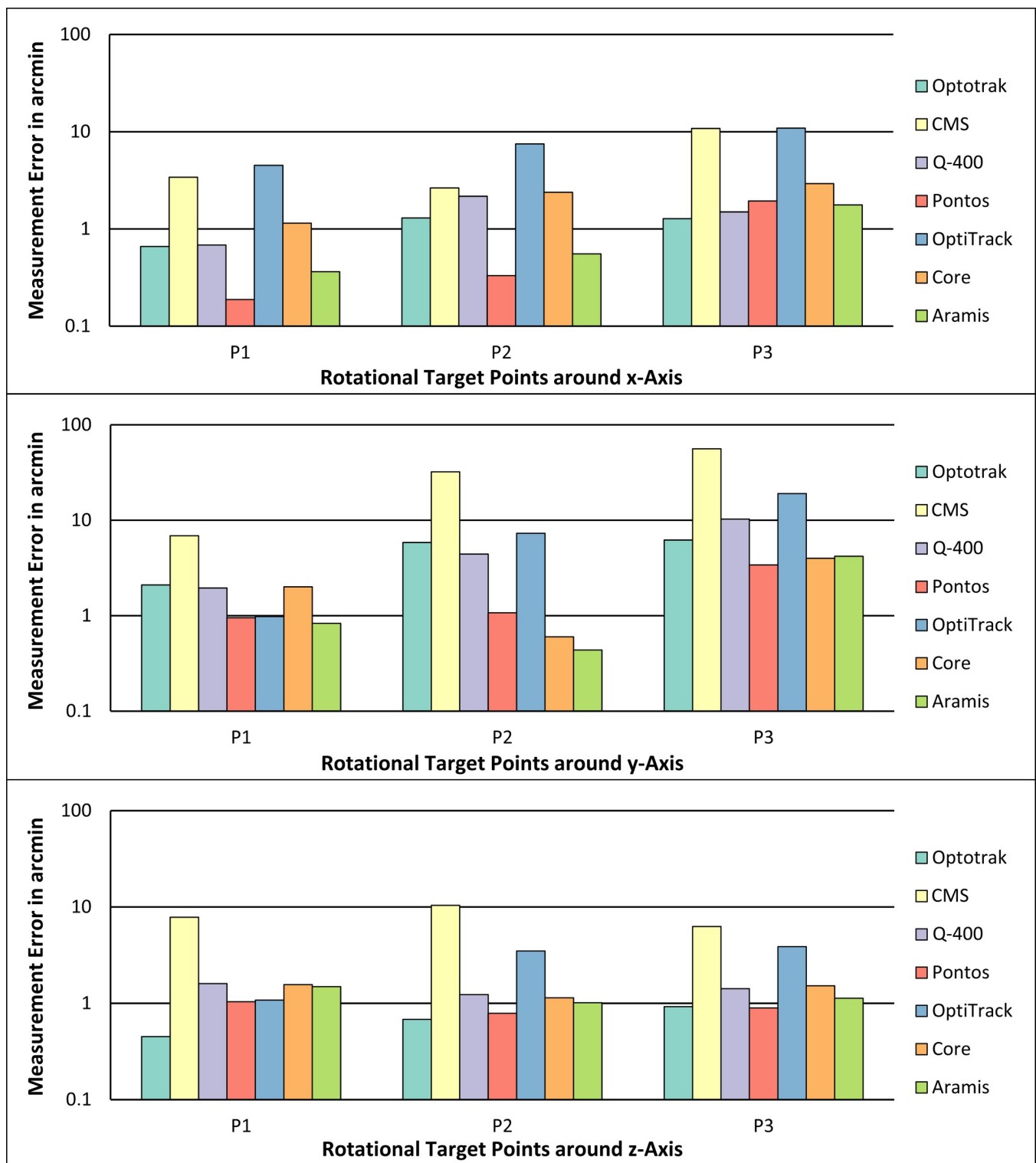

**Fig 7.** Measurement errors of the accuracy measurements of the translational motions in x-direction (top diagram), y-direction (middle diagram) and z-direction (bottom diagram) for the seven measurement systems using the manual reference device.

**Table 7. Numerical rotational measurement errors of the seven measurement systems when using the manual reference device.**

|  | Optotrak | CMS | Q-400 | Pontos | OptiTrack | Core | Aramis |
|---|---|---|---|---|---|---|---|
| P1_X in arcmin | 0.66 ± 0.42 | 3.40 ± 2.07 | 0.68 ± 0.55 | 0.19 ± 0.12 | 4.50 ± 0.22 | 1.15 ± 0.46 | 0.36 ± 0.45 |
| P2_X in arcmin | 1.30 ± 0.65 | 2.64 ± 1.96 | 2.17 ± 0.52 | 0.33 ± 0.25 | 7.48 ± 0.21 | 2.38 ± 0.41 | 0.56 ± 0.44 |
| P3_X in arcmin | 1.27 ± 0.63 | 10.79 ± 1.98 | 1.50 ± 0.91 | 1.94 ± 0.21 | 10.88 ± 0.28 | 2.93 ± 0.35 | 1.77 ± 0.75 |
| P1_Y in arcmin | 2.10 ± 0.77 | 6.89 ± 6.92 | 1.95 ± 1.07 | 0.95 ± 0.29 | 0.98 ± 0.52 | 2.00 ± 0.45 | 1.12 ± 0.51 |
| P2_Y in arcmin | 5.84 ± 1.42 | 32.16 ± 9.67 | 4.42 ± 2.06 | 1.07 ± 0.27 | 7.31 ± 0.73 | 0.60 ± 0.20 | 0.68 ± 0.37 |
| P3_Y in arcmin | 6.20 ± 1.12 | 56.17 ± 12.78 | 10.31 ± 3.19 | 3.40 ± 0.36 | 19.07 ± 2.32 | 4.00 ± 0.23 | 4.51 ± 0.56 |
| P1_Z in arcmin | 0.45 ± 0.25 | 7.82 ± 7.29 | 1.60 ± 0.25 | 1.04 ± 0.36 | 1.08 ± 0.39 | 1.57 ± 0.39 | 1.49 ± 0.14 |
| P2_Z in arcmin | 0.68 ± 0.37 | 10.39 ± 4.02 | 1.23 ± 0.24 | 0.79 ± 0.33 | 3.50 ± 0.68 | 1.14 ± 0.33 | 1.02 ± 0.27 |
| P3_Z in arcmin | 0.92 ± 0.35 | 6.26 ± 3.15 | 1.42 ± 0.24 | 0.90 ± 0.43 | 3.88 ± 0.58 | 1.52 ± 0.43 | 1.13 ± 0.22 |

However, only translational motions can be applied because the CMM only possesses linear motors. The largest movements can be performed in x-direction followed by the z-direction followed by the y-direction. The maximum translation of all axes was determined using the maximum motion of the y-direction. The manual reference device enables the comparison of translational and rotational motions between different measurement systems. However, due to the necessity of a user to adjust the different rotational and translational motions, this system is more prone to individual errors. In addition, only small translational motions can be applied due to the limit of the micrometer screws. For users of camera based motion capture systems who don't have any high precision reference systems like the CMM, the manual reference device can be a rational and relatively inexpensive option. The translation in x-, y- and z-direction of P4, using the CMM, and P3, using the manual reference system, were both 5 mm in total. Therefore, the measurement errors P4 of the CMM and P3 of the manual device can be compared. For the Optotrak, Pontos, Core and Aramis systems slightly higher errors were detected in the measurements using the manual system in comparison with the CMM. For the OptiTrack system it is the other way round. For the CMS and Q-400 system, the CMM at P4 provoke slightly higher measurement error compared to P3 of the manual device in x-direction, contrarily to the y- and z-directions. However, marker attachment, the distance of the measurement systems to the markers or other environmental factors may have a higher impact on the measurement errors than the reference system or the evaluation algorithm.

## Comparison of the different measurement systems

In general, the two measurement systems using active markers showed the highest measurement errors for both reference devices. Comparing these two measurement systems, the Optotrak revealed better measurement accuracies than the CMS system. In the case of the Optotrak, the measurement did not take place in the optimal measurement volume calibrated by the manufacturer. The measurement in the calibrated measurement volume was not possible due to the size of the air-conditioned measurement room. The highest measurement errors for the passive markers systems were seen for the OptiTrack system. The Q-400 system showed good results for all motions, especially for small movements up to 5 mm. The OptiTrack system used seven and the Q-400 system used three cameras. The accuracy could decrease or increase by reducing or adding cameras, depending on the requirements [4]. The best results for the translational and rotational motions were seen for the three stereo camera systems (Pontos, Core and Aramis), which were all from the same manufacturer and use 2D reference point markers. A comparison of different motion capture systems was published by Richards in 1999 using a self-constructed motorized device to change the passive maker positions [15].

Topley and Richards recently repeated the measurements to compare the measurement accuracy of modern optoelectronic motion capture systems with the systems 20 years earlier [16]. They showed that the development of modern motion capture systems led to an advanced measurement accuracy. Compared to the stereo camera measurement systems within this study, the measurements by Topley and Richards revealed much higher measurement errors. However, they used measurement systems and spherical passive markers which are typically used for human motion capture and the distances between the markers were much greater than in the current test. Topley and Richards used a digitizer to determine the positions of the passive markers having an absolute accuracy of 0.036 mm. For the small measurement errors of the measurement systems in the current study, which are partly below 20 μm or even below 10 μm, an even more precise reference system like the manual adjustment unit and the CMM were needed. The measurement systems in the study by Topley and Richards were typically used for human motion capture analysis, where marker positioning and soft tissue motion provoke higher measurement errors than the systems analyzed [16]. One important parameter next to the accuracy is the maximum analyzable volume. A larger analyzable volume leads to a lower resolution and therefore, to a lower accuracy. As shown in Table 1, the active marker systems and the two measurement systems consisting of multiple cameras cover a larger volume than the stereo camera systems. The volumes of the stereo camera systems can be adjusted by using a bigger calibration plate, but only to a limited extend. Hence, the stereo camera systems are not recommended for human motion capture analysis, where large joint motions should be analyzed. Also a speckle pattern, as used by the Q-400 system, seems to be unsuitable for human motion capture analysis, due to the low depth measurement. However, in the current study, the maximum distance for determining the measurement accuracy was in an area of 100 mm, which is much smaller than the distances during human motion analyses. The main goal was to determine the accuracy for biomechanical measurement, like cadaver joint motions and implant fixation, which need a high accuracy. For analysis of cadaver joint motions, all measurement systems of the current study seem to be useful, but this depends on the extent of the motion. For bigger joint motions the systems, which cover a larger volume are more useful. Typically, the rigid body markers systems offer the user the possibility to select bony landmarks and define an anatomic coordinate system, which is a useful tool for measurement of joint kinematics. However, for biomechanical high precision measurements like micro-motions between implant and bone or measurements of the fracture gaps of bones supplied with osteosynthesis plates under loading conditions, where small volumes are sufficient, the passive stereo camera systems of the current study are of mature interest. Depending on the biomechanical application, the maximum frequency of the measurement system is an important factor as well.

## Limitations

The study design has some limitations, which need to be addressed. First, general statements about the measurement accuracy cannot be made, as each measurement system is represented only once. Although every laboratory in the round robin test is a frequent user of their measurement system, results could differ when the calibration, measurement and data analysis with the same measurement system is performed by another user. In the current study the data analysis (moving marker to motionless marker for the CMM and moving marker to the same marker at zero position for the manual reference device) were identical between the groups.

Another limitation is the relatively small size of the measurement room (17 m$^2$ floor space and 2.5 m height). For the Optotrak system which covers a big volume the optimum distance

between marker and measurement system was not achieved. Therefore, the measurement accuracy of this system is expected to be higher if the minimum distance is given. For the other measuring systems the minimum distance could be guaranteed.

Ultimately, the CMM seems to be a reliable tool for reference measurements of different measurement systems. It has a similar reproducibility and no user could accidentally influence its motions. However, only translational motions can be driven with the CMM.

The manual system seems to be a simple and useful tool to compare the translational and rotational measurement accuracies of different measurement systems. To account for interpersonal difference one user performed the adjustments of the manual device for every group.

Problematic for the determination of the rotatory accuracies around the different axes was the existence of different rotatory units with different accuracies. This means that a comparison of the measurement errors between the three axes of a measuring system is not meaningful. However, a comparison of the different systems regarding the rotational accuracy can be made. The complete tests were run under optimized conditions and the markers were placed on solid bodies. Therefore, the absolute values of measurement errors determined in this study cannot be transferred directly into a biomechanical application, where the markers are typically fixated on soft tissue or bones.

## Conclusions

The results of the current study show advantages regarding the measurement accuracy of single motions up to 100 mm for the measurement systems using passive markers and especially for the stereo camera systems. However, depending on the requirements of the user and the application, other factors like the measurement frequency, the maximum analyzable volume, the marker type and the costs for the measurement system are important factors to be considered.

## Supporting information

**S1 Table. CMM data.** Measurement data from the seven measurement systems using the setup with the coordinate measurement machine. Values of the zero offset measurement and for the measurement points approached in the x, y and z directions.
(XLSX)

**S2 Table. Manual reference device data.** Measurement data from the seven measurement systems using the setup with the manual reference device. Values for the measuring points approached in the x, y and z directions and the rotations performed about these three axes.
(XLSX)

## Acknowledgments

The authors would like to thank the MSB-Net for organizing the yearly meeting of biomechanical laboratories for the development and implementation of new project ideas. The current study emerged from the MSB-Net meeting.

## Author Contributions

**Conceptualization:** Bastian Welke.

**Data curation:** Sebastian Jaeger, Jonas Schwer, Andreas Martin Seitz, Isabell Hamann, Michael Werner, Christoph Thorwaechter, Inês Santos, Toni Wendler, Dennis Nebel.

**Formal analysis:** Stefan Schroeder, Sebastian Jaeger, Jonas Schwer, Andreas Martin Seitz, Isabell Hamann, Michael Werner, Christoph Thorwaechter, Inês Santos, Toni Wendler, Dennis Nebel.

**Investigation:** Stefan Schroeder, Sebastian Jaeger, Jonas Schwer, Andreas Martin Seitz, Isabell Hamann, Michael Werner, Christoph Thorwaechter, Inês Santos, Toni Wendler, Dennis Nebel, Bastian Welke.

**Methodology:** Stefan Schroeder, Bastian Welke.

**Supervision:** Bastian Welke.

**Visualization:** Stefan Schroeder.

**Writing – original draft:** Stefan Schroeder.

**Writing – review & editing:** Sebastian Jaeger, Jonas Schwer, Andreas Martin Seitz, Isabell Hamann, Michael Werner, Christoph Thorwaechter, Inês Santos, Toni Wendler, Dennis Nebel, Bastian Welke.

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
