## [Decision Letter · Decision Letter 0]

10 May 2022

PONE-D-22-08505Accuracy measurement of different marker based motion analysis systems for biomechanical applications: A round robin studyPLOS ONE

Dear Dr. Welke,

Thank you for submitting your manuscript to PLOS ONE. After careful consideration, we feel that it has merit but does not fully meet PLOS ONE’s publication criteria as it currently stands. Therefore, we invite you to submit a revised version of the manuscript that addresses the points raised during the review process. Main points of revision provided by the reviewers are related to the necessity of more discussion and providing more data on the experimental setup.  

We look forward to receiving your revised manuscript.

Kind regards,

Antonio Riveiro Rodríguez, PhD

Academic Editor

PLOS ONE

Journal Requirements: When submitting your revision, we need you to address these additional requirements. 1. Please ensure that your manuscript meets PLOS ONE's style requirements, including those for file naming. The PLOS ONE style templates can be found at https://journals.plos.org/plosone/s/file?id=wjVg/PLOSOne_formatting_sample_main_body.pdf and https://journals.plos.org/plosone/s/file?id=ba62/PLOSOne_formatting_sample_title_authors_affiliations.pdf. 2. We note that the grant information you provided in the ‘Funding Information’ and ‘Financial Disclosure’ sections do not match.  When you resubmit, please ensure that you provide the correct grant numbers for the awards you received for your study in the ‘Funding Information’ section. 3. Acknowledgments Section: Move New Information to the Financial Disclosure:"Thank you for stating the following in the Acknowledgments Section of your manuscript: [The authors would like to thank the MSB-Net for organizing the yearly meeting of biomechanical laboratories for the development and implementation of new project ideas. The current study emerged from the MSB-Net meeting. We acknowledge support by the German Research Foundation (DFG) and the Open Access Publication Fund of Hannover Medical School (MHH).] We note that you have provided funding information that is not currently declared in your Funding Statement. However, funding information should not appear in the Acknowledgments section or other areas of your manuscript. We will only publish funding information present in the Funding Statement section of the online submission form. Please remove any funding-related text from the manuscript and let us know how you would like to update your Funding Statement. Currently, your Funding Statement reads as follows:  [The author(s) received no specific funding for this work.Publication costs are covered by the German Research Foundation (DFG) and the Open Access Publication Fund of Hannover Medical School (MHH).] Please include your amended statements within your cover letter; we will change the online submission form on your behalf. 4. Thank you for stating the following in the Competing Interests section: [I have read the journal's policy and the authors of this manuscript have the following competing interests: [Sebastian Jaeger reports grants from B Braun Aesculap, Johnson & Johnson Depuy Synthes, Heraeus Medical, Waldemar Link, Peter Brehm, Ceramtec, Implantcast, Mathys Orthopaedie GmbH and Zimmer Biomet that are not related to the current study]
 .  Please confirm that this does not alter your adherence to all PLOS ONE policies on sharing data and materials, by including the following statement: "This does not alter our adherence to  PLOS ONE policies on sharing data and materials.” (as detailed online in our guide for authors http://journals.plos.org/plosone/s/competing-interests).  If there are restrictions on sharing of data and/or materials, please state these. Please note that we cannot proceed with consideration of your article until this information has been declared.  Please include your updated Competing Interests statement in your cover letter; we will change the online submission form on your behalf. 5. Please amend either the abstract on the online submission form (via Edit Submission) or the abstract in the manuscript so that they are identical. 6. Please include a caption for figure 4.  7. Please review your reference list to ensure that it is complete and correct. If you have cited papers that have been retracted, please include the rationale for doing so in the manuscript text, or remove these references and replace them with relevant current references. Any changes to the reference list should be mentioned in the rebuttal letter that accompanies your revised manuscript. If you need to cite a retracted article, indicate the article’s retracted status in the References list and also include a citation and full reference for the retraction notice.

Reviewers' comments:

Reviewer's Responses to Questions

**Comments to the Author**

1. Is the manuscript technically sound, and do the data support the conclusions?

Reviewer #1: Yes

Reviewer #2: Yes

2. Has the statistical analysis been performed appropriately and rigorously? 

Reviewer #1: Yes

Reviewer #2: Yes

3. Have the authors made all data underlying the findings in their manuscript fully available?

Reviewer #1: Yes

Reviewer #2: Yes

4. Is the manuscript presented in an intelligible fashion and written in standard English?

Reviewer #1: Yes

Reviewer #2: Yes

5. Review Comments to the Author

Reviewer #1: Authors presented the comparison between different camera-based systems for movement analysis. In general, the paper is well-written and presents important information for researchers in biomechanics and instrumentation. However, some clarifications are needed.

- More discussion on each test should be included, the setups should be clearer presented as well as the positioning of each measurement system.

- Limit of detection of each technology should be clarified.

- More discussion of the preferable system for each test should be included.

- There is an error in the figure numbering.

Reviewer #2: Review of „Accuracy measurement of different marker-based motion analysis systems for biomechanical applications: A round robin study”

This paper is a review and at the same time a benchmarking article for different marker-based motion analysis system. The main importance of this paper is to provide objective accuracy measurement, which is beneficial for researchers who are about to purchase new equipment for laboratory or existing laboratories to refer to solid error measurements for their works. Authors present clear and reliable study with good presentation of background and results. This work is written in a device-oriented manner, whereas in the title there are biomechanical applications. I wish to present some issues that could contribute to further improvement of this work.

1. From method section in abstract, you pointed out that study was performed in one room. In method section line 97-100 you said that laboratories were tested. I think that it requires clarification, if by laboratories you meant equipment or there were different rooms in each laboratory.

2. This study could provide more information about measurement circumstances. In 2nd section you also stated that the room was temperature controlled. For purpose of making this study replicable, I think that you should add information about exact temperature in the room if someone wish to repeat your setup.

3. In line 71 you mentioned that skin and general circumstances of marker attachment makes it prone to measurement errors. I cannot find in description if you tried to mimic the body tissue or markers were put on solid surface of the device. If so, this could also be mentioned as limitation of the study.

4. From now on I would focus on your limitations, or rather lack of specificity of setup descriptions. You lack specifics in terms of this issues. Line 356 – data analysis could be user-specific. You mentioned that some computations were made for device designed software, but you do not share exact computation methods for each device. Is it irrelevant or you based on raw results? If algorithms could vary, for the same sake of repeatability, you should supplement that information.

5. If you state that size of the room could affect measurements – please provide the size data for this room. Line 358.

6. Line 362 – what do you mean by slightly larger distance? Could you please provide “ideal” conditions for measurement setup? For example, manufacturer recommendations?

7. I think that discussion also lacks with reference to actual biomechanical measurements. For human movement, test for movement with 10 cm range is good for pelvis movement during walking or hand manipulation. I suspect that with increase of movement speed and gross motor analysis like swings or kicks errors could be much greater. Could you please extrapolate results or makes this issue as another limitation, based on your own judgement? One sentence at the end of paper is not satisfying.

In my opinion this paper can be publish after minor revision, focusing on providing more experiment setup data for scientific community.

I wish you good luck with further work.

6. PLOS authors have the option to publish the peer review history of their article (what does this mean?). If published, this will include your full peer review and any attached files.

Reviewer #1: No

Reviewer #2: **Yes: **Dariusz Mosler

---

## [Author Response · Author response to Decision Letter 0]

14 Jun 2022

Antonio Riveiro Rodríguez, PhD

Academic Editor

PLOS ONE

Journal Requirements: 

Thank you very much for this hint. We have changed it accordingly.

3. Acknowledgments Section: Move New Information to the Financial Disclosure:

"Thank you for stating the following in the Acknowledgments Section of your manuscript: 

[The authors would like to thank the MSB-Net for organizing the yearly meeting of biomechanical laboratories for the development and implementation of new project ideas. The current study emerged from the MSB-Net meeting. We acknowledge support by the German Research Foundation (DFG) and the Open Access Publication Fund of Hannover Medical School (MHH).]

 [The author(s) received no specific funding for this work.

Publication costs are covered by the German Research Foundation (DFG) and the Open Access Publication Fund of Hannover Medical School (MHH).]

[I have read the journal's policy and the authors of this manuscript have the following competing interests: [Sebastian Jaeger reports grants from B Braun Aesculap, Johnson & Johnson Depuy Synthes, Heraeus Medical, Waldemar Link, Peter Brehm, Ceramtec, Implantcast, Mathys Orthopaedie GmbH and Zimmer Biomet that are not related to the current study]

. 

Thank you very much for this hint. We have changed it accordingly.

6. Please include a caption for figure 4. 

 Thank you for pointing this out. We have added the caption to the manuscript.

 We have thoroughly checked the references again

Reviewers' comments:

Reviewer's Responses to Questions 

Comments to the Author

1. Is the manuscript technically sound, and do the data support the conclusions?

Reviewer #1: Yes

Reviewer #2: Yes

2. Has the statistical analysis been performed appropriately and rigorously? 

Reviewer #1: Yes

Reviewer #2: Yes

3. Have the authors made all data underlying the findings in their manuscript fully available?

Reviewer #1: Yes

Reviewer #2: Yes

4. Is the manuscript presented in an intelligible fashion and written in standard English?

Reviewer #1: Yes

Reviewer #2: Yes

5. Review Comments to the Author

Reviewer #1: Authors presented the comparison between different camera-based systems for movement analysis. In general, the paper is well-written and presents important information for researchers in biomechanics and instrumentation. However, some clarifications are needed.

Thanks a lot for your positive feedback. All information (line numbers) on the changes refer to the document "Revised Manuscript with Track changes".

- More discussion on each test should be included, the setups should be clearer presented as well as the positioning of each measurement system.

Thank you for your suggestion. We added the following parts to the method section:

Line 130 and 132: “…is positioned in the corner of the measurement room and has an accuracy of 2.0 µm. The size of the measurement room is 17 m2 floor space and 2.5 m height.”

Line 136: “Two adapters, having a flat rectangular surface,…”

Line 148 – 151: “Thus, for each axis, six measurements were performed at zero position and five measurements were performed at a distance of 10 mm. The distances from the moving adapter were determined in relation to the markers of the motionless adapter and compared to the real motion of the CMM to calculate the measurement errors. “

Line 152-153: “…the same approach was used as for the previous measurement, but…”

Line 157-158: “After each position, a measurement was performed. Using this method, the measurement errors could be determined from very low motions (0.1 mm) to larger distances (100 mm).”

Line 170-172: “The manual reference device was fixated at a stable metal frame, which was positioned next to the CMM in the same measurement room. The different markers were fixated on the front plate of the reference device.”

Line 173-174: “Therefore, no reference markers on any motionless part were needed.”

Line 179: “Measurement were taken at each of the indicated positions.”

Furthermore, the following parts were implemented to the discussion section:

Line 314-325: “The CMM is a user independent and reproducible device, which makes it a suitable tool to determine measurement errors of different measurement systems. However, only translational motions can be applied because the CMM only possesses linear motors. The largest movements can be performed in x-direction followed by the z-direction followed by the y-direction. The maximum translation of all axes was determined using the maximum motion of the y-direction. The manual reference device enables the comparison of translational and rotational motions between different measurement systems. However, due to the necessity of a user to adjust the different rotational and translational motions, this system is more prone to individual errors. In addition, only small translational motions can be applied due to the limit of the micrometer screws. For users of camera based motion capture systems who don’t have any high precision reference systems like the CMM, the manual reference device can be a rational and relatively inexpensive option.”

- Limit of detection of each technology should be clarified.

Thank you very much. We already stated the detection limit of the system according to the producer’s declarations in Table 1 within the point “Measurement Accuracy”. In addition, the study should help to get the “real” accuracy limits. 

Next to the accuracy further limits were found in Tab. 1 like the maximum analyzable volume and the maximum measurement frequency. We added possible applications for each measurement system to the discussion part with the focus on the specific limits.

Line 368-380: However, in the current study, the maximum distance for determining the measurement accuracy was in an area of 100 mm, which is much smaller than the distances during human motion analyses. The main goal was to determine the accuracy for biomechanical measurement, like cadaver joint motions and implant fixation, which need a high accuracy. For analysis of cadaver joint motions, all measurement systems of the current study seem to be useful, but this depends on the extent of the motion. For bigger joint motions the systems, which cover a larger volume are more useful. Typically, the rigid body markers systems offer the user the possibility to select bony landmarks and define an anatomic coordinate system, which is a useful tool for measurement of joint kinematics. Depending on the biomechanical application, the maximum frequency of the measurement system is an important factor as well.

- More discussion of the preferable system for each test should be included.

Thanks for this comment. We tried to sum up the results in the discussion part and describe the differences between the systems regarding the accuracy and possible biomechanical applications. A detailed discussion of the different systems could, in our view, lead to an emphasis on certain measurement systems, which we do not intend to do due to our required neutrality. We tried to expand the discussion as followed:

Line 286-289: The three stereo camera systems and the CMS system did not show an increasing measurement error with an increasing distance. These results show that the stereo camera systems had the highest measurement accuracies for the zero offset and translational motions up to 100 mm. For longer distances however, no statements can be made. 

Line 301-303: Thus, for very small translational motions, the three stereo camera systems and the Q-400 System showed the lowest measurement errors. 

Line 307-309: For biomechanical applications, where typically translations and rotations occur, the translational and rotational measurement errors could sum up, which may lead in even higher total measurement errors.

- There is an error in the figure numbering.

You are right, we changed it accordingly.

Reviewer #2: Review of „Accuracy measurement of different marker-based motion analysis systems for biomechanical applications: A round robin study”

This paper is a review and at the same time a benchmarking article for different marker-based motion analysis system. The main importance of this paper is to provide objective accuracy measurement, which is beneficial for researchers who are about to purchase new equipment for laboratory or existing laboratories to refer to solid error measurements for their works. Authors present clear and reliable study with good presentation of background and results. This work is written in a device-oriented manner, whereas in the title there are biomechanical applications. I wish to present some issues that could contribute to further improvement of this work.

Thank you very much for your positive feedback. All information (line numbers) on the changes refer to the document "Revised Manuscript with Track changes".

1. From method section in abstract, you pointed out that study was performed in one room. In method section line 97-100 you said that laboratories were tested. I think that it requires clarification, if by laboratories you meant equipment or there were different rooms in each laboratory.

Thanks for the hint that was not clear. In fact, all measurements were performed in the same room in the same laboratory. Different measurement systems of different laboratories were tested.

We changed the Sentence in Line 98-99 accordingly: “All measurements were performed in the same standardized temperature-controlled (22 ± 1°C) precision measurement room.“

2. This study could provide more information about measurement circumstances. In 2nd section you also stated that the room was temperature controlled. For purpose of making this study replicable, I think that you should add information about exact temperature in the room if someone wish to repeat your setup.

Good point, we added the information of the room temperature accordingly.

Line 99: (22 ± 1°C)

3. In line 71 you mentioned that skin and general circumstances of marker attachment makes it prone to measurement errors. I cannot find in description if you tried to mimic the body tissue or markers were put on solid surface of the device. If so, this could also be mentioned as limitation of the study.

For the current study we wanted to compare the measurement accuracy of different motion capture systems under optimum conditions. Therefore, we placed the markers on solid bodies. 

We added the following part to the limitation section:

Line 406-409: “The complete tests were run under optimized conditions and the markers were placed on solid bodies. Therefore, the absolute values of measurement errors determined in this study cannot be transferred directly into a biomechanical application, where the markers are typically fixated on soft tissue or bones.”

4. From now on I would focus on your limitations, or rather lack of specificity of setup descriptions. You lack specifics in terms of this issues. Line 356 – data analysis could be user-specific. You mentioned that some computations were made for device designed software, but you do not share exact computation methods for each device. Is it irrelevant or you based on raw results? If algorithms could vary, for the same sake of repeatability, you should supplement that information.

Yes, good point. In general, the data analysis influences the results. For good group comparison, we specified the type of data analysis. We added the following to the limitation section:

Line 387-389: In the current study the data analysis (moving marker to motionless marker for the CMM and moving marker to the same marker at zero position for the manual reference device) were identical between the groups.

5. If you state that size of the room could affect measurements – please provide the size data for this room. Line 358.

Yes, we added the size of the room.

Line 131-132 & 390: (17 m2 floor space and 2.5 m height).

6. Line 362 – what do you mean by slightly larger distance? Could you please provide “ideal” conditions for measurement setup? For example, manufacturer recommendations?#

The slightly larger distance was more like a feeling from the experience of the user. The Co-authors found some information about the necessary distance between measurement system and object, which was 80-100 cm. Therefore, the optimum distance was given for the OptiTrack system and we deleted the following text accordingly: “For the OptiTrack system, there is no specification for a minimum distance, but from the experience of the users, a slightly larger distance would probably also lead to better results.”

7. I think that discussion also lacks with reference to actual biomechanical measurements. For human movement, test for movement with 10 cm range is good for pelvis movement during walking or hand manipulation. I suspect that with increase of movement speed and gross motor analysis like swings or kicks errors could be much greater. Could you please extrapolate results or makes this issue as another limitation, based on your own judgement? One sentence at the end of paper is not satisfying.

Thank you for this comment. You are right, the range of 100 mm is too small for typical human motion analyses. In our round robin tests we focused on biomechanical applications, which need high accuracies and have small ranges. These are mainly cadaver joint and implant fixation analyses. We believe that for human motion analyses during gait or sport activities accuracies below 100 µm are not absolutely necessary, which of course also depends on the respective research question. We added the following to the discussion part:

Line 368-380: However, in the current study, the maximum distance for determining the measurement accuracy was in an area of 100 mm, which is much smaller than the distances during human motion analyses. The main goal was to determine the accuracy for biomechanical measurement, like cadaver joint motions and implant fixation, which need a high accuracy. For analysis of cadaver joint motions, all measurement systems of the current study seem to be useful, but this depends on the extent of the motion. For bigger joint motions the systems, which cover a larger volume are more useful. Typically, the rigid body markers systems offer the user the possibility to select bony landmarks and define an anatomic coordinate system, which is a useful tool for measurement of joint kinematics. However, for biomechanical high precision measurements like micro-motions between implant and bone or measurements of the fracture gaps of bones supplied with osteosynthesis plates under loading conditions, where small volumes are sufficient, the passive stereo camera systems of the current study are of mature interest.

And furthermore, depending on the biomechanical research question, the maximum frequency of each measurement system is also an important factor. We added the following:

Line 379-380: Depending on the biomechanical application, the maximum frequency of the measurement system is an important factor as well.

In my opinion this paper can be publish after minor revision, focusing on providing more experiment setup data for scientific community.

I wish you good luck with further work.

6. PLOS authors have the option to publish the peer review history of their article (what does this mean?). If published, this will include your full peer review and any attached files.

Do you want your identity to be public for this peer review? For information about this choice, including consent withdrawal, please see our Privacy Policy.

Reviewer #1: No

Reviewer #2: Yes: Dariusz Mosler

---

## [Decision Letter · Decision Letter 1]

29 Jun 2022

Accuracy measurement of different marker based motion analysis systems for biomechanical applications: A round robin study

PONE-D-22-08505R1

Dear Dr. Welke,

We’re pleased to inform you that your manuscript has been judged scientifically suitable for publication and will be formally accepted for publication once it meets all outstanding technical requirements.

Kind regards,

Antonio Riveiro Rodríguez, PhD

Academic Editor

PLOS ONE

Reviewers' comments:

Reviewer's Responses to Questions

**Comments to the Author**

1. If the authors have adequately addressed your comments raised in a previous round of review and you feel that this manuscript is now acceptable for publication, you may indicate that here to bypass the “Comments to the Author” section, enter your conflict of interest statement in the “Confidential to Editor” section, and submit your "Accept" recommendation.

Reviewer #1: All comments have been addressed

Reviewer #2: All comments have been addressed

2. Is the manuscript technically sound, and do the data support the conclusions?

Reviewer #1: Yes

Reviewer #2: Yes

3. Has the statistical analysis been performed appropriately and rigorously? 

Reviewer #1: Yes

Reviewer #2: Yes

4. Have the authors made all data underlying the findings in their manuscript fully available?

Reviewer #1: Yes

Reviewer #2: Yes

5. Is the manuscript presented in an intelligible fashion and written in standard English?

Reviewer #1: Yes

Reviewer #2: Yes

6. Review Comments to the Author

Reviewer #1: Authors addressed my comments and suggestions. I recommend the publication of this paper in the current format

Reviewer #2: As all issues have been adressed in a proper manner, and I indicate that only minor changes should be done, I recommend publishing this paper. Good luck for the author and I am looking formward to another, more extensive studies of this kind from your team.

7. PLOS authors have the option to publish the peer review history of their article (what does this mean?). If published, this will include your full peer review and any attached files.

Reviewer #1: No

Reviewer #2: **Yes: **Dariusz Mosler

---

## [Editor Report · Acceptance letter]

1 Jul 2022

PONE-D-22-08505R1 

Accuracy measurement of different marker based motion analysis systems for biomechanical applications: A round robin study 

Dear Dr. Welke:

I'm pleased to inform you that your manuscript has been deemed suitable for publication in PLOS ONE. Congratulations! Your manuscript is now with our production department. 

Kind regards, 

on behalf of

Dr. Antonio Riveiro Rodríguez 

Academic Editor

PLOS ONE